# An Application of Inverse Reinforcement Learning to Estimate Interference in Drone Swarms

**DOI:** 10.3390/e24101364

**Published:** 2022-09-27

**Authors:** Keum Joo Kim, Eugene Santos, Hien Nguyen, Shawn Pieper

**Affiliations:** 1Thayer Engineering School, Dartmouth College, 15 Thayer Drive, Hanover, NH 03755, USA; 2Computer Science Department, University of Wisconsin-Whitewater, McGraw Room 106, 800 W. Main Street, Whitewater, WI 53190, USA

**Keywords:** drone swarms, entropy, interference, inverse reinforcement learning, reward distributions, homogeneity, correlation analysis

## Abstract

Despite the increasing applications, demands, and capabilities of drones, in practice they have only limited autonomy for accomplishing complex missions, resulting in slow and vulnerable operations and difficulty adapting to dynamic environments. To lessen these weaknesses, we present a computational framework for deducing the original intent of drone swarms by monitoring their movements. We focus on interference, a phenomenon that is not initially anticipated by drones but results in complicated operations due to its significant impact on performance and its challenging nature. We infer interference from predictability by first applying various machine learning methods, including deep learning, and then computing entropy to compare against interference. Our computational framework begins by building a set of computational models called double transition models from the drone movements and revealing reward distributions using inverse reinforcement learning. These reward distributions are then used to compute the entropy and interference across a variety of drone scenarios specified by combining multiple combat strategies and command styles. Our analysis confirmed that drone scenarios experienced more interference, higher performance, and higher entropy as they became more heterogeneous. However, the direction of interference (positive vs. negative) was more dependent on combinations of combat strategies and command styles than homogeneity.

## 1. Introduction

Drone technologies for intelligently coordinating various behaviors across diverse situations [1,2,3,4] become more advanced every day, which enhances their ability to perform complicated and sensitive missions, especially as the drones increase in size [5,6,7]. Escalating demands require higher levels of drone autonomy in which individual drones can make their own decisions more freely and collaboratively [3]. In practice, however, drones still have only limited autonomy to accomplish missions provided by skilled human supervisors, resulting in operations that are slow, vulnerable, and inflexible at adapting to dynamic environments.

To address these shortcomings, we present a computational framework to deduce the original intent of drone swarms by monitoring their movements. To do so, we focus on interference, defined as a phenomenon unanticipated by drones and resulting from complicated operations by multiple drones, since interference has been identified as a key to understanding those challenges but is not yet well-formulated. We present here an approach to infer interference from predictability obtained by the learning methods. In addition, entropy is computed to compare against interference while analyzing various drone scenarios. We apply inverse reinforcement learning (IRL) to identify reward distributions that best explain sequences of drone movements and assume these reward distributions represent the preferences of individual decision-makers. The entropy and interference computed based on these reward distributions were used to deduce the original intents of drone controllers.

In this paper, we analyze a variety of drone simulation scenarios generating discernible differences in performance, entropy, and interference. Our objective is to validate the hypothesis that drone swarms faced with high interference behave less predictably than those with low interference when utilizing IRL to identify reward distributions used to compute both entropy and interference. With the help of the computed entropy and interference, we were able to answer the following questions: (1) Can interference be used to differentiate groups of varying homogeneity? (2) What were the performances and entropies of those groups? (3) How is interference different from entropy? and (4) Is interference positively or negatively correlated with performance?

To answer these questions, all scenarios of interest were grouped based on homogeneity of combat strategies and command cognitive styles in ally and enemy teams of drones. By comparing these groups of scenarios, we found that scenarios with greater heterogeneity performed better and had both higher entropy and higher interference than those with less heterogeneity. Although both entropy and interference were lowest in the most homogeneous group and highest in the most heterogeneous group, they were different in those groups where only one of command style and combat strategy was homogeneous while the other was heterogeneous. In the group where only combat strategy was homogeneous, the interference was higher than in the other group where only command style was homogeneous. On the other hand, the entropy was higher in the group where only command style was homogeneous than in the group where only combat strategy was homogeneous.

This paper is organized as follows: First, we describe the methods used in our research, including (1) a double transition model (DTM) of an applied computational model, (2) inverse reinforcement learning (IRL) of an algorithmic approach to reveal reward distributions from trajectories of drones, (3) entropy and interference for analysis of drone swarms, and (4) a description of the simulation system developed for this analysis. Next, we analyze the results of simulating various drone scenarios. Lastly, we present our discussions and conclusions.

## 2. Materials and Methods

To analyze drone swarms by computing entropy and interference based on reward distributions driven by inverse reinforcement learning (IRL), we made use of a computational model (i.e., Double Transition Model) as explained in Section 2.1, developed an IRL algorithm as described in Section 2.2, computed entropy and interference as formulated in Section 2.3, and simulated drone swarms as provided in Section 2.4. A drone swarm consists of 10 drones with five drones on each side (i.e., an ally side and an enemy side) [8]. Drones on the same side work cooperatively, competing against drones on the opposing side. DTMs were built using the behavioral trajectories of 100 repeated simulations of 10 agents’ combat scenarios. Consequently, 100 behavioral trajectories were utilized for each DTM.

Figure 1 describes our workflow to analyze drone scenarios using reward distributions revealed by IRL. Each drone scenario was investigated by 100 DTMs built from 50/50 ally and enemy trajectories of drones. The reward distributions were obtained by applying IRL to DTMs and utilized to estimate entropy and interference of drone scenarios.

### 2.1. Double Transition Model (DTM)

The double transition model (DTM) [9] is a computational framework developed to describe the process of human opinion formation and change. One of the basic assumptions of DTM is that decisions are a series of actions taken by individuals, as described by the MDP (Markov decision process) [10]. The MDP formulates the state transition, under Markov property, that the next state is determined only by the current state and the action taken at that time and is independent of all other previous states and actions, while addressing its utility scheme appropriately. Based on MDP, the decision making process can be modeled as DTM, and the reward distributions can be computed by IRL (inverse reinforcement learning).

DTMs treat human opinion as a cognitive state, and the process of opinion formation and change is encoded by sequential transitions of cognitive states. An example of the human decision-making process (i.e., a decision trajectory) can then be represented by traversing nodes of cognitive states, where DTMs are demonstrated by graphs composed of nodes of cognitive states and edges of transitions between these states. Assuming two types of cognitive states, we considered two types of nodes: one for query transition and the other for memory transition. The interconnection between these two types of nodes can contribute to general cognitive transition, as human cognitive states are associated with memory or query depending on the circumstances. Previously, DTMs were validated and applied for representing human decision-making processes [11]. In this paper, we construct multiple DTMs where each DTM can be varied by a combination of four combat strategies and three cognitive styles of decision-makers.

Figure 2 shows one of the DTMs built and analyzed as an MDP. (A) shows the DTM, which contains 3925 nodes (states) and 4751 edges (actions), in its entirety, while (B) shows a part of the DTM containing a collection of states from 324 to 343 and actions described as edges. The combat strategy of Force Field and command style of spontaneous were employed for both enemy and ally sides of this drone swarm.

### 2.2. Inverse Reinforcement Learning (IRL)

We reveal reward distributions that best explain the sequence of decisions using inverse reinforcement learning (IRL [12]) based on DTMs built from behavior records. A set of decision sequences of a drone swarm can be modeled as DTMs and represented by alternating cognitive states and actions as described below:(1)$$\tau =\left\{s_{1},a_{1},s_{2},\ldots ,\;s_{k-1},a_{k-1},s_{k}\right\}$$

By assuming that a set of sequences of decisions belonging to a swarm provides a representative sampling of decisions of the swarm in the environment given, we make use of the frequency ratio of the combinations of state and action observed in the behavioral records to address the behavioral pattern of the swarm. This frequency ratio is encoded as triple rewards and linear expected rewards (LERs), defined as below and used in our analysis to compute entropy and interference.
(2)$$LER\left(\tau \right)=\overset{K-1}{\underset{i=1}{\sum }}P_{w}\left(s_{i+1}|a_{i},s_{i}\right)r\left(s_{i},a_{i},s_{i+1}\right)$$
where K denotes the length of trajectory τ. The triple reward $r\left(s_{i},a_{i},s_{i+1}\right),$ computed from IRL, is expected to represent differences among drones, and $P_{w}$ is expected to capture the (internal) biases of a drone *w*.

In each DTM, we considered two groups of ally and enemy: 50 ally side decision trajectories were set for $T_{1}$ and 50 decision trajectories for enemy side were set for $T_{2}$. By defining upper and lower bounds on the LER of these groups of trajectories as below:(3)$$lb\left(T_{i}\right)=\underset{\tau \in T_{i}}{\min}LER\left(\tau \right)$$
(4)$$ub\left(T_{i}\right)=\underset{\tau \in T_{i}}{\max}LER\left(\tau \right)$$
we enforce a partial ordering over these groups as following:(5)$$\delta \left(T_{i},T_{i+1}\right)=lb\left(T_{l}\right)-ub\left(T_{l+1}\right)\geq 1$$

Reward triples, $r\left(s,a,s^{\prime }\right),$ are inferred as:(6)$$r\left(s,a,s^{\prime }\right)=peak*P_{T_{1}}\left(a|s\right)+\Delta r_{T_{1}}\left(s,a,s^{\prime }\right)\;$$
(7)$$r\left(s,a,s^{\prime }\right)=-peak*P_{T_{2}}\left(a|s\right)+\Delta r_{T_{2}}\left(s,a,s^{\prime }\right)$$
where peak is a predefined reward cap (e.g., 100). $P_{T_{1}}\left(a|s\right)$ reflects the ally’s preference of selecting an action in a given state based on the frequency ratio computed from the original behavioral records (similarly, $P_{T_{2}}\left(a|s\right)$ reflects the enemy’s preference). $\Delta r_{T_{1}}\left(s,a,s^{\prime }\right)$ and $\Delta r_{T_{2}}\left(s,a,s^{\prime }\right)$ are fractional reward values that capture the variance in rewards of each side.

Note that if a triple (i.e., $\left(s,a,s^{\prime }\right)$) is shared by both the ally and the enemy sides, both Equations (6) and (7) apply, forcing the fractional rewards to compromise.

In our IRL, the objective is set to minimize the overall fractional reward variance by minimizing the magnitude of the largest fraction reward. Therefore, both a collection of trajectories and LERs formulated by Equations (1) and (2) would represent the preference of drone swarms. Based on these assumptions, we utilize LERs to compute entropy and to infer indirect interference of drone swarms.

### 2.3. Entropy and Interference

In this section, we briefly review entropy and interference and explain how they were computed using reward distributions.

#### 2.3.1. Entropy

Entropy, which stems from thermodynamics, can be used to understand the behavior of stochastic processes since it represents the uncertainty, ambiguity, and disorder of the process without placing additional restrictions on the theoretical probability distribution [13]. It has previously been studied in planning drone searches for objects [14] and in analyzing drone robustness [15,16]. In a study by Cofta et al. [15], drone behaviors were treated as Shannon stochastic information sources, and their communication signals were used to compute entropy. In another study by Reddy et al. [16], entropy was computed based on convex-hull size and the number of paths to secure and locate the original drone signals while avoiding adversarial attacks. In addition, entropy was used to estimate the volume of useful information acquired per time for planning drone paths [14]. In this paper, we compute the entropy formulated by Shannon entropy theory [17] to estimate the uncertainty of drone swarms, by utilizing LERs in Equation (2) as below:(8)$$H\left(X\right)=-\overset{n}{\underset{i=1}{\sum }}P\left(x_{i}\right)\log P\left(x_{i}\right)$$
where *X* is a set of LERs revealed by IRL, and *P*$\left(x_{i}\right)$ denotes the probability of $x_{i}$ occurring. We consider LERs for both the ally and enemy sides to compute the overall entropy of the drone swarm.

#### 2.3.2. Interference

Unlike entropy, interference has yet to be well-formulated [18]. Interference leads to observed values differing from actual values, and it often refers to unanticipated signals to anticipated signals. Previously, interference has been estimated in a study by Candel et al. [19], where the signal-to-interference ratio of the communication channel within a robotic arm force-seeking scenario was predicted by a random forest method. In this prior work, interference was considered to distort the anticipated signals in a disruptive manner and to harm performance. This negative interference has been investigated by Kumar et al. [20], Li and Kim [21], and Uddin et al. [22]. 

However, in another study, some interference has been proven to be beneficial to overall performance, since it reduces the probability of another interference occurring [23]. In a study by Dou et al. [24], both positive and negative interference were experimentally observed. According to the simulation results provided by Ran and Chen [25], external information interference greatly promoted the dissemination of original information in social networks. From the human genome project [23] to social network study [25], interference has influenced performance both positively and negatively. It is likely that interference impacts performance negatively initially up to a certain limit, but becomes beneficial beyond the limit. Otherwise, the impact of interference depends on the environmental condition or source of the interference. This opposing argument or dual effect of interference motivated us to study interference while analyzing drone swarms using reward distributions. In drone swarms simulated for our study, drone interaction and autonomy were controlled indirectly by specifying combat strategy and command style.

Due to the complex and challenging nature of interference, we relied on predictability to infer interference rather than exact measurements. The predictability was determined by learning methods applied to reward distributions revealed by IRL [26,27] which were based on DTMs [9]. We hypothesized that drone swarms are easily predictable if they experience no interference, but that they are hardly predictable otherwise, and we validated this hypothesis against various drone scenarios.

A scenario of drone swarm (D) is described by a four-tuple (ACMD, ECMD, AS, ES), where:ACMD is an ally side cognitive style of commanderECMD is an enemy side cognitive style of commanderAS is an ally side combat strategyES is an enemy side combat strategy

Based on the above hypothesis, we formulated the interference as below:(9)y ∝f(g(D))
where *g*(*D*) denotes the predictive accuracy of a scenario (*D*), and *y* is the interference to be inferred.

### 2.4. Drone Swarm Simulation

We developed a system of simulating drone swarms using MA-Gym [28], a collection of multi-agent environments based on Open-AI Gym [29], and we applied the default configuration combat environment in MA-Gym. A variety of combat scenarios were implemented by specifying the combat strategy and cognitive style of the commander. The four combat strategies of Clear the Swarm (CS), Birds of a Feather (BF), Force Field (FF) and All or Nothing (AN), and the three cognitive styles of rational, spontaneous, and intuitive were implemented.

Combat strategies are implemented as different prioritizations of behaviors: CS prioritizes eliminating enemy agents, BF prioritizes staying near ally agents, FF prioritizes staying near the ally ship, and AN prioritizes eliminating the enemy ship, and they were implemented as function calls initiated at the beginning of each scenario. For the cognitive style, rational style drones stay close to their allies and ship, and only attack enemies if they are very close to each other; spontaneous drones stray further from their allies and ship and attack enemies that are further away, and intuitive drones demonstrate behaviour somewhere in the middle, between the two. Command styles are implemented as a series of distances that would affect the strategies to be executed, with rational reflecting more cautious behavior, spontaneous more reckless, and intuitive somewhere in between (e.g., a rational drone would stay closer to allies than a spontaneous drone in BF, and a spontaneous drone would go to enemies that are farther away more readily than a rational drone in CS). Drone behaviors are determined by both combat strategy and cognitive style. These combat strategies and command styles can be changed anytime during the simulation since they are implemented as functions applicable at every timestep. Once a swarm begins execution, it continues until either all the ships of one side are destroyed or the maximum time allowed is reached. All trajectories representing agents’ behaviors are collected for analysis.

A behavioral trajectory consists of a sequence of features representing the environment and action taken at each timestep. Features include each agent’s health, ammo, 2D Cartesian coordinates (in ranges of four grid spaces), and discretized Euclidean distance to the nearest ally, enemy, terrain, and their own ship, as well as the ship’s health (0–5). An agent can take one of four types of actions: move on the grid, hover in place, attack, or start. Moving on the grid was recorded as one of (“UP”, “DOWN”, “LEFT”, “RIGHT”), hovering as (“NOOP”), attacking as (“ATTACK”), and starting as (“START”) taken at the beginning of the game.

## 3. Results

We simulated 144 combat scenarios in which swarms were composed of two sides, ally and enemy, and each side was composed of five drones. Each scenario was specified by combining the three command styles of Intuitive, Rational, and Spontaneous, as well as the four combat strategies of CS, BF, FF, and AN for each side. Swarms of each scenario were simulated 100 times repeatedly with randomization, and behavioral trajectories were gathered and used for building DTMs.

### 3.1. Performance

We used the number of games won by the ally side in 100 repetitions of each scenario as a measure of performance. Winning or losing the game is determined by whether either a team’s ship is destroyed or all drones in a team died. If either of those was not accomplished within the given time limit (i.e., 50 steps), whoever has the most *(health + (ammo/2))* wins the game. Throughout this paper, we compared z-scores of performance values standardized by Equation (10):(10)$$Z=\frac{X-E\left(X\right)}{\sigma \left(X\right)}$$
where E(X) denotes the mean of *X* and σ(X) denotes the standard deviation of *X*.

Figure 3 shows the z-score computed from the performance measured based on simulations performing each scenario. The x-axis denotes the index of scenarios specified by combat strategy and cognitive style of commanders of both sides, and the y-axis represents the z-score of the performance measured from each scenario. As shown in Figure 1, the scenario was identified as a deterministic factor for performance. 64 out of 144 scenarios were found to have a positive z-score, while 80 out of 144 scenarios were found to have a negative z-score. The overall mean z-score was −0.0303 with a standard deviation of 0.9627; the minimum and maximum z-scores were −1.5115 and 2.3889, respectively. The mean of 64 positive z-scores was 0.778 with a standard deviation of 0.7626, and the mean of 80 negative z-scores was −0.6770 with a standard deviation of 0.5116. The overall distribution of performance was skewed slightly positive (i.e., 80/64), but its negative portion was spread more widely compared to its positive portion.

### 3.2. Entropy

Based on DTMs developed from drone behavioral trajectories, we collected LERs in Equation (2) revealed by IRL and computed entropies associated with a variety of drone scenarios. For each scenario, 100 DTMs were constructed in which each DTM relied on 50/50 ally/enemy trajectories resulting in 100 LERs, and 100 entropy values were calculated as Equation (8). By dealing with swarm behaviors as sources of information, we anticipated the entropy computed here to represent the amount of information enclosed in each DTM. We compared the z-scores of the entropy values standardized by Equation (10).

Figure 4 shows the heatmap plotting z-scores of the entropy computed from 14,400 simulations. A total of 85 of the 144 scenarios were found to have positive z-scores, while 59 of the 144 scenarios were found to have negative z-scores. The overall mean z-score was 9.251 × 10^−18^ with the standard deviation of 0.973, the minimum z-score was −1.525, and the maximum z-score was 2.933. The mean of 85 positive z-scores was 0.599 with a standard deviation of 0.719, and the mean of 59 negative z-scores was −0.863 with a standard deviation of 0.539. Similarly to the performance result, the z-score distribution of entropy was skewed slightly positive (i.e., 85/59). The scenarios with negative z-scores were more narrowly clustered around the mean than the scenarios with positive z-scores. Figure 4 reveals that the scenarios generated a discernible difference in entropy.

### 3.3. Interference

We inferred interference from the predictive accuracy obtained by machine learning methods while predicting parameters of scenarios using reward distributions revealed by IRL. A drone swarm scenario (D) was specified by (ACMD, ECMD, AS, ES) as explained in Section 2.3. To estimate the interference associated with these parameters, we conducted experiments in which a parameter of interest is predicted by machine learning methods. A reward distribution was composed of a set of LERs and computed by applying IRL to DTMs based on behavioral trajectories of drones involved in a swarm as defined by Equation (9). Since a swarm simulation was conducted by 50/50 ally/enemy trajectories, each reward distribution was composed of 100 numeric values. Therefore, a sample was composed of 100 feature values (i.e., LERs) and labeled by a parameter of interest during learning.

Predicting ACMD and ECMD, the labels of rational, intuitive, or spontaneous were applied to represent the commander cognitive styles of ally or enemy side. The labels of Clear the Swarm (CS), Birds of Feather (BF), Force Field (FF), and All or Nothing (AN) for combat strategies were used for predicting AS and ES.

Figure 5 shows the predictive accuracy obtained by applying RF (Random Forest) [30] and SVM (Support Vector Machine) [31] for predicting parameters specifying drone scenarios utilizing reward distributions uncovered by IRL. (A) shows the accuracy obtained by RF, and (B) provides the accuracy obtained by SVM. We utilized 14,400 samples for learning each parameter, and measured the accuracy. The experiment was conducted by 10-fold cross-validation and repeated three times. The x-axis denotes the fold index from 1 to 30, and the y-axis represents a parameter of interest for each experiment. Each cell corresponds to the accuracy obtained by each fold of learning. The lighter the cell color, the higher the accuracy achieved for that fold.

Regarding the topmost heatmap obtained by RF, the overall mean accuracy was 0.563 with a standard deviation of 0.2037. For each parameter, the mean and standard deviation of accuracies accomplished by predicting ES, AS, ECMD, and ACMD were 0.770 ± 0.015, 0.703 ± 0.015, 0.3876 ± 0.015, and 0.3872 ± 0.023, respectively. Regarding the heatmap on the bottom obtained by SVM, the overall mean of accuracy was 0.523 with a standard deviation of 0.1750. For each parameter, the mean and standard deviation of accuracies accomplished by predicting ES, AS, ECMD, and ACMD were 0.710 ± 0.018, 0.634 ± 0.020, 0.375 ± 0.015, and 0.372 ± 0.015, respectively.

Naturally, the accuracies varied depending on the learning method and parameter of interest selected. RF outperformed SVM in all cases. However, the predictive accuracy of ES was highest, and that of ACMD was the lowest in both methods. This result supports our assumption that predictive accuracy would be a key to estimating interference. In addition, we found that strategies of ES and AS were more predictable than command cognitive styles. The differences between strategies (i.e., ES and AS) and between command cognitive styles (i.e., ECMD and ACMD) were much smaller than the differences between combat strategy and command cognitive style (i.e., ES and ECMD, AS, and ACME) in the same or opposite side. We interpret this result as evidence that command cognitive style is associated more with interference than combat strategies. The different numbers of classes for command cognitive style (i.e., 3) and combat strategy (i.e., 4) can be further evidence to support the predictive accuracy as a key to estimating interference.

#### 3.3.1. Variations by Learning Method

Predictive accuracy should depend on the learning method and data selected. How different would predictive accuracy be if we change the learning method? To answer this question, in this section, we explored various learning approaches.

##### Machine Learning

Machine learning has been developed to assist computers in improving their performance. It builds a mathematical model using sample data and makes decisions on new data without being explicitly programmed. In recent years, machine learning evolved into deep learning approaches, which are more powerful in dealing with massive amounts of data. The key advances of deep learning methods are driven by the accumulation of massive amounts of data and the highly powerful parallel computing of GPUs [32]. For machine learning, we applied Adaboost [33], Decision Tree [34], Naive Bayes [35] and QDA (Quadratic Discriminant Analysis) [34] to the same dataset and examined their variations. For deep learning, we varied the optimizer (i.e., Adam [36], RMSprop [37], Nadam [38], and Ftrl [39]) and activation function (i.e., relu, selu, sigmoid, softmax, softplus, and tanh) to examine their consistency in predictability.

Figure 6 shows the predictive accuracies obtained from Adaboost, Decision Tree, Naive Bayes, and QDA to predict parameters of interest using LERs. (A) shows the accuracy obtained by Adaboost where the overall minimum and maximum accuracies were 0.333 and 0.608, while their mean and standard deviations were 0.463 and 0.102, respectively. The minimum and maximum of accuracies predicting ES, AS, ECMD, and ACMD were [0.488: 0.608], [0.510: 0.592], [0.349: 0.410], and [0.333: 0.406], where those means and standard deviations were 0.544 ± 0.029, 0.557 ± 0.020, 0.383 ± 0.014, and 0.366 ± 0.019, respectively. (B) shows the accuracy obtained by the decision tree, where the overall minimum and maximum were 0.3208 and 0.6138, while their mean and standard deviation were 0.4548 and 0.1102. The minimum and maximum accuracies predicting ES, AS, ECMD and ACMD were [0.558: 0.614], [0.465: 0.549], [0.338: 0.414], and [0.321: 0.394], where those means and standard deviations were 0.583 ± 0.016, 0.511 ± 0.018, 0.37 ± 0.020, and 0.356 ± 0.016, respectively. (C) shows the accuracy obtained by Naive Bayes, where the overall minimum and maximum were 0.283 and 0.696, while their mean and standard deviation were 0.461 and 0.152. The minimum and maximum accuracies predicting ES, AS, ECMD, and ACMD were [0.614: 0.696], [0.474: 0.557], [0.283: 0.369], and [0.318: 0.379], where those means and standard deviations were 0.652 ± 0.021, 0.515 ± 0.023, 0.332 ± 0.019, and 0.347 ± 0.016, respectively. (D) shows the accuracy obtained by QDA, where the overall minimum and maximum were 0.344 and 0.686, while their mean and standard deviation were 0.496 and 0.137. The minimum and maximum accuracies predicting ES, AS, ECMD, and ACMD were [0.625: 0.688], [0.532: 0.604], [0.344: 0.430], and [0.349: 0.413], where those means and standard deviations were 0.650 ± 0.018, 0.571 ± 0.020, 0.386 ± 0.018, and 0.375 ± 0.017, respectively. For all learning methods, we conducted 10-fold cross-validation and repeated it three times. Therefore, each cell in Figure 4 represents a predictive accuracy obtained once by learning.

F or all five learning methods we tested (i.e., RF, SVM, Adaboost, Decision Tree, Naive Bayes, and QDA), the predictive accuracy was higher when predicting the strategy than when predicting the command style. For most cases, the enemy side was more predictable than the ally side for both combat strategy and command style, although there were some exceptions. For instance, the ally side strategy was more predictable than the enemy strategy for the case of AdaBoost, and the ally command style was more predictable than the enemy command style in Naive Bayes. Although there are minor variations depending on the learning method applied, the predictive accuracy was determined mainly by the parameter of interest, which motivated us to explore the predictive accuracy as a key to estimate interference. Observing that the accuracy associated with strategy was higher than that associated with command style, we assumed that the interference associated with command style would contribute to low predictive accuracy. Although various learning methods can be utilized to infer interference, a consistent pattern of higher interference associated with command style rather than combat strategy is anticipated. For implementing multiple machine learning methods, we used the Scikit-learn library [40].

##### Deep Learning

We built several deep learning models with various optimizers and activation functions and examined how much their accuracies would vary depending on our parameters of interest. The activation functions were relu, selu, sigmoid, softmax, softplus, and tanh, and the optimizers were Adam, RMSprop, Nadam, and Ftrl. Our learning model was composed of input, output, and two hidden layers. The input layer contained 100 feature values obtained from reward distributions obtained by DTMs. The first hidden layer contained 50 nodes, the second hidden layer was composed of 25 nodes, and the output layer was mapped into our parameter of interest. For ES and AS, four combat strategies were used for output and three command styles were used for ECMD and ACMD.

Figure 7 shows the predictive accuracy obtained by applying deep learning composed of two hidden layers in addition to input and output layers. The learning rate was set to 0.001. Each cell represents an average accuracy obtained by the algorithm and activation function specified. (A) shows the results achieved by Adam optimization, a stochastic gradient descent method based on the adaptive estimation of first-order and second-order moments. With activation functions of relu, selu, sigmoid, softmax, softplus, and tanh as noted in the x-axis, we obtained the accuracies of 0.856, 0.856, 0.797, and 0.796. (B) provides the accuracy obtained by RMSprop that maintains the moving (discounted) average of the square of gradients. With activation functions of relu, selu, sigmoid, softmax, softplus, and tanh as noted by the x-axis, we achieved accuracies of 0.856, 0.856, 0.797, and 0.797 for ES, AS, ECMD, and ACMD, respectively. (C) provides the accuracy obtained by Nadam, a variant of Adam with Nesterov momentum. With activation of relu, selu, sigmoid, softmax, softplus, and tanh, we obtained the accuracies of 0.854, 0.855, 0.797, and 0.797 for ES, AS, ECMD, and ACMD, respectively. (D) provides the accuracy obtained by Ftrl (Follow the Regularized Leader) developed by Google for large and sparse feature spaces. With activation of relu, selu, sigmoid, softmax, softplus, and tanh, we obtained the accuracies of 0.854, 0.856, 0.796, and 0.797. For building deep models, we used the TensorFlow library [41].

Like machine learning, for all combinations of optimizer and activation functions, we observed a consistent pattern that the predictive accuracy was determined mainly by either the combat strategy or the command style. This supports our key assumption that the interference associated with the command style would contribute to low predictive accuracy. As anticipated, the accuracies obtained by deep learning were higher than those obtained by machine learning, and the gaps between combat strategy and command style became smaller by deep learning. In addition, the gaps between enemy and ally sides became negligible by deep learning, although those gaps were considerable with machine learning.

Although deep learning is a promising technique to consider, our research objective here is to validate our hypothesis to infer interference from predictability rather than to identify the best learning method. Therefore, we present a few ways to infer interference from predictive accuracy in the next section. As a measure of predictive accuracy obtained by all learning models under consideration, we used *F*1-score [42], which combines the precision and recall as formulated in Equation (11):(11)$$F1\;Score=\frac{2*\left(Precision*Recall\right)}{\left(Precision+Recall\right)}$$

#### 3.3.2. From Predictive Accuracy to Interference

To infer interference from predictive accuracy, we applied four mathematical models, presented in Table 1. To examine the fitness of a model for interference, we conducted the experiment with all 144 classes of scenarios. For our research convenience, we applied RF for all following experiments. The mean and standard deviation (i.e., STD) of accuracy were 0.075 and 0.010, respectively, and the minimum and maximum accuracies were 0.054 and 0.093, respectively.

The first column in Table 1 denotes an index addressed by Figure 8, and range shows the minimum and maximum interference derived from accuracy *x* when each model is applied. We also reported the mean and STD. Since plot (B) best manifested interference for the same accuracies under our consideration, we next chose it as our interference model and explored it with respect to homogeneity.

We note that the cap of the interference should be specified to get reasonable interference values from predictive accuracy. In some cases, the infinity of interference can be derived from a predictive accuracy close to zero. In these cases, maximum interference would be more informative than infinity.

### 3.4. Homogeneity vs. Interference

By assuming homogeneity and heterogeneity to be associated with interference by nature, in this section, we divided all scenarios under consideration into two groups, one for homogeneous scenarios and the other for heterogeneous scenarios, and estimated the interference of each group.

#### 3.4.1. Homogeneity in Combat Strategy

All scenarios were partitioned into two groups: one group (G1) of scenarios with the same strategy for both sides, and the other group (G2) of scenarios where the strategies of the two sides were set differently. We applied RF to predict parameters of interest using reward distributions uncovered by IRL. In Figure 7, the two heatmaps of (A) and (B) show the predictive accuracies and the two heatmaps of (C) and (D) show the interference derived by applying model B in Table 1 to the accuracies obtained by RF. The mean and standard deviation of G1 were 0.670 and 0.331, respectively, while those of G2 were 0.552 and 0.278, respectively. The minimum and maximum accuracies of G1 were 0.278 and 0.994, respectively, while those of G2 were 0.343 and 0.774, respectively. By observing that the accuracies of G1 were higher than those of G2, we confirmed that the homogeneous scenarios were more predictable than the heterogeneous scenarios.

By applying the model B in Table 1, we inferred the interference as shown by (C) and (D) in the bottom of Figure 9. The mean and standard deviation of the interference of G1 were 1.836 and 0.915, respectively, while those of G2 were 2.000 and 0.703, respectively. The minimum and maximum interference of G1 were 1.006 and 3.600, respectively, while those of G2 were 1.292 and 2.919, respectively. As expected, the interference of G1 was lower than that of G2.

#### 3.4.2. Homogeneity in Command Style

Analogously, we partitioned all scenarios into two groups depending on the homogeneity of command cognitive style: one group (G3) of scenarios set with the same command style on both sides, and the other group (G4) of scenarios set with different command styles on each side. Figure 10 shows the accuracy obtained by RF and the interference derived. The mean and standard deviation of G3 were 0.577 and 0.159, respectively, while those of G4 were 0.558 and 0.197, respectively, as shown by (A) and (B). The minimum and maximum accuracies of G3 were 0.367 and 0.817, respectively, while those of G4 were 0.344 and 0.819, respectively. By comparing these accuracies, we confirmed that the accuracies of G3 were higher than those of G4. The interferences derived from the accuracies were plotted by two heatmaps of (C) and (D). The mean and standard deviation of the interference of G3 were 1.843 and 0.501, respectively, while those of G4 were 1.980 and 0.694, respectively. The minimum and maximum interferences of G3 were 1.224 and 2.727, respectively, while those of G4 were 1.221 and 2.909, respectively. These results confirmed that the interference of G3 was lower than that of G4.

### 3.5. Scenarios by Homogeneity

In this section, we partitioned all scenarios into four subgroups where both combat strategy and command style were the same on both sides (H1), only combat strategies were the same (H2), only command cognitive styles were the same (H3), and neither combat strategy nor command style were the same (H4). The level of homogeneity of H1 is the highest, while that of H4 is the lowest. Comparing scenarios by homogeneity showed that the group of highest interference performed best. However, this cannot guarantee that drone scenarios experiencing high interference always perform well or not, since our experiments were limited to comparing scenarios by homogeneity and to generating diverse homogeneity.

In addition, positive, and negative interferences found by correlation analysis revealed that the direction of interference was more relevant to combination of combat strategy and command style than homogeneity. Homogeneity is a key factor to interference, but limited in impacting performance directly.

#### 3.5.1. Performance

Figure 11 shows the performance of these groups of scenarios by homogeneity. The *x*-axis represents the index from 1 to 100 to denote 100 simulations conducted for each scenario, and the y-axis denotes the scenario for groups of H1, H2, H3, and H4. The mean performances were −0.211, −0.180, 0.057, and 0.067, with standard deviations of 0.682, 0.624, 1.017, and 1.003, for H1, H2, H3, and H4, respectively. The mean z-score of performance improved as the scenarios became more heterogeneous.

#### 3.5.2. Entropy

Figure 12 shows the entropy of these groups of scenarios by homogeneity. The x-axis represents the random index from 1 to 100, and the y-axis denotes the scenario for each group. The mean entropy was −0.279, 0.001, 0.029, and 0.032, with standard deviations of 1.427, 1.442, 0.791, and 0.772 for H1, H2, H3, and H4 respectively. The mean z-score of entropy increased as the scenarios became more heterogeneous.

#### 3.5.3. Interference

We inferred the interference based on the predictive accuracy as shown by Figure 13. The RF learning method was applied, and each cell in the heatmaps corresponds to an interference value based on a predictive accuracy accomplished by learning once. The mean interferences were 2.495, 5.624, 3.790, and 7.663, with standard deviations of 0.337, 1.292, 0.420, and 1.035 for H1, H2, H3, and H4, respectively. The interference inferred from H1 was much lower than that from H4. The increasing heterogeneity of groups of scenarios increased the interference.

As anticipated by the previous analysis, the interference of H2 (5.624) was higher than that of H3 (3.790) as shown by Figure 13, which implies that command cognitive style impacts interference more than combat strategy. However, the entropy and performance of H3 were higher than those of H2, which reveals that the interference was more sensitive to heterogeneous command cognitive styles, unlike the entropy and performance.

In summary, we discovered that the most heterogeneous group of scenarios performed best and obtained the highest entropy and interference. This result implies that we can regulate the interference by homogeneity or estimate homogeneity by interference. However, this finding cannot support positive or negative correlations between performance and interference. Since the dual effect of interference is of research interest, as addressed by Dou et al. [24] and Ran and Chen [25], we performed correlation analysis of homogeneity with scenarios in each group.

### 3.6. Correlation Analysis

So far, we compared the scenarios grouped by homogeneity and found the most heterogeneous group of scenarios to have the highest performance, interference, and entropy. However, this finding neither explains the correlations among those metrics nor addresses the dual effect of interference. Therefore, we performed the correlation analysis with all scenarios under consideration.

The overall Pearson correlation between performance and interference was 0.046 with a *p*-value of 0.582. The correlation between performance and entropy was 0.365 with a *p*-value of 6.920 × 10^−6^. The correlation between interference and entropy was 0.010 with a *p*-value of 0.901. Only the correlation between performance and entropy seemed statistically meaningful, but its strength of 0.365 was too weak to form any conclusions. This might be since positive and negative correlations coexist across a variety of drone scenarios. Therefore, we performed additional correlation analysis with scenarios by group of homogeneity while separating them into positive and negative subgroups.

Figure 14 shows the correlations among performance, entropy, and interference in each positive and negative subgroup. The plots from (A) to (D) show the correlations in positive subgroups, and the plots from (E) to (H) show the correlations in negative subgroups. Table 2 presents the *p*-values and numbers of scenarios associated with each correlation value displayed by Figure 14. Table 3 provides the combinations of combat strategy and command style identified, associated with positive and negative correlations.

As demonstrated by (A) H1(+), we obtained 0.7 correlation between performance and interference from the positive subgroup of H1. However, this was revealed to be statistically insignificant since the *p*-value was 0.185, as shown in Table 2. As shown by (B) H2(+), the correlation of 0.75 was computed with the *p*-value of 0.005 from the positive subgroup of H2. As shown by (C) H3(+), the performance and interference were correlated with 0.51 of *p*-value 0.029 in the positive subgroup of H3. As shown by (D) H4(+), the performance and interference were correlated with 0.53 of *p*-value 0.001 in the positive subgroup of H4. In summary, strong correlations between performance and interference were found in all positive subgroups with varying *p*-values.

As shown by (E) H1(−), the correlations of 0.94 between performance and entropy, −0.88 between performance and interference, and −0.74 between interference and entropy were found to have *p*-values of 0.002, 0.008, and 0.056, respectively, from the negative subgroup of H1 as provided by Table 2. Two of these correlations were significant statistically (i.e., *p*-value < 0.01). As shown by (F) H2(−), the correlations of 0.93 between performance and entropy and −0.57 between performance and interference were found with *p*-values of < 0.001 and 0.055, respectively, from the negative subgroup of H2. As shown by (G) H3(−), the correlations of 0.48 between performance and entropy and −0.74 between performance and interference with *p*-values of 0.042 and < 0.001, respectively, were found from the negative subgroup of H3. As shown by (H) H4(−), the correlations of 0.58 between performance and entropy and −0.54 between performance and interference with *p*-values of <0.001 and 0.001, respectively, were found from the negative subgroup of H4.

Comparing the top four heatmaps against the bottom four heatmaps describing positive vs. negative interferences, it is noticeable that the correlations between performance and entropy in (E), (F), (G), and (H) were 0.94 (*p* = 0.002), 0.93 (*p* < 0.001), 0.485 (*p* = 0.042), and 0.576 (*p* < 0.001), which were higher than 0.24 (*p* = 0.702), −0.11(*p* = 0.726), 0.44 (*p* = 0.068), and 0.40 (*p* = 0.011), as shown in (A), (B), (C), and (D). This result shows high correlations between performance and entropy under negative interference.

Table 2 provides the *p*-values and numbers of scenarios associated with correlations provided by Figure 12. The left part shows positive correlations, and the right part shows negative correlations. The first column ‘H’ denotes the group homogeneity ID as explained in Section 3.5; the correlation values in the column ‘correlation’ were computed by applying ‘metric1′ and ‘metric2′ to the Pearson method. The value of ‘# positive’ denotes the number of drone scenarios having positive correlations, while ‘# negative’ denotes the number of scenarios having negative correlations. We found that the interference was rather negatively correlated with performance when considering both correlation strength and statistical significance together. Regarding positive correlations between performance and interference, H1, H2, H3, and H4 showed the correlations of 0.704 (*p* = 0.185), 0.745 (*p* = 0.005), 0.514 (*p* = 0.029), and 0.527 (*p* = 0.001), respectively. Regarding negative correlations, H1, H2, H3, and H4 showed correlations of −0.884 (*p* = 0.008), −0.567 (*p* = 0.055), −0.744 (*p*< 0.001), and −0.538 (*p* = 0.001), respectively. For the most homogeneous group H1, more scenarios were found to have negative interference, while more scenarios in the most heterogeneous group H4 were found to have positive interference.

Table 3 presents combinations of combat strategy and command style contributing to positive and negative correlations between performance and interference involved in each group of scenarios. The column ‘H’ denotes the index of group homogeneity explained in Section 3.5, while ‘st1’ and ‘st2’ denote the strategies specified for enemy and ally sides, respectively. Similarly, ‘style1’ and ‘style2’ denote the command styles specified for enemy and ally sides, respectively. The combat strategies were denoted by CS for Clear the Swarm, BF for Birds of a Feather, FF for Force Field and AN for All or Nothing while the command styles were described by INT for intuitive, RATNL for rational and SPONT for spontaneous styles. From this analysis, we found the homogeneity was not a major factor in determining the direction of interference. With respect to the most homogeneous group H1, the interference became positive when the strategy BF was combined with the spontaneous command style, but it became negative when BF was combined with other command styles, such as intuitive and rational, as shown by Table 3. Interestingly, the strategy CS was involved in negative interference only, regardless of the command style to be utilized regarding H1. We assumed combinations of combat strategy and command style produced confounding factors, so that the interference impacted performance variously.

## 4. Discussion

We seek to develop a computational framework to deduce the original intent of drone controllers by monitoring drone movements. This intent can be described as discovering how drones are formed in terms of strategy taken and which decision-making style is employed by Commander. Consequently, these factors affect performance, entropy, and interference, as shown by the analysis performed. As noted, group homogeneity was specified by combat strategy and commander style, and performance was measured by the number of games in which the ally side won over 100 game repetitions.

It is not surprising that the most heterogeneous scenarios performed best. Previously, researchers have pointed out that group diversity improves team performance because people with different interests and ways of acting may prevent the group from being stuck in a problem-solving process. Our results showed that is the case with a group of drones as well, by showing that the most heterogenous group obtained the highest interference and performance. However, we were limited in generating heterogenous drones guided by different Commander styles with different strategies and could not fully examine interference, entropy, and performance within the same homogeneous group of scenarios. Our team is developing more combat strategies to further extend this research.

This paper contributes a computational framework to understand decision-maker intent by monitoring sequences of decisions. By tracking the movements of drones, we collected their records, built DTMs, uncovered reward distributions, and estimated interference and entropy. In addition, we identified whether interference impacted performance positively and under which combinations of combat strategies and command styles. Although limited to our simulations, we accomplished our objective by estimating interference, a key to understanding the original intent of drone controllers, using reward distributions. This capability can be bounded by drone monitoring technology and simulation capability.

## 5. Conclusions

In this paper, we demonstrated our attempt to analyze drone swarms using inverse reinforcement learning (IRL). By computing and analyzing the entropy and interference using reward distributions uncovered by IRL, we found that as scenarios became more heterogeneous, they performed better, experienced greater interference, and resulted in higher entropy. In addition, the entropy and interference showed different patterns among scenarios where either combat strategy or command style was heterogeneous while the other was homogeneous. Finally, by investigating the dual effect of interference through correlation analysis across a variety of drone scenarios, we identified combinations of combat strategy and command cognitive style contributing to interferences positively and negatively.

In summary, our approach can be utilized to analyze drone swarms using IRL, to regulate entropy or interference by homogeneity, and to identify combat strategy and command cognitive style by interference. Although our current analysis is limited to combat strategies and command styles under consideration, this approach is generic and applicable to diverse domains.

## Figures and Tables

**Figure 1 entropy-24-01364-f001:**
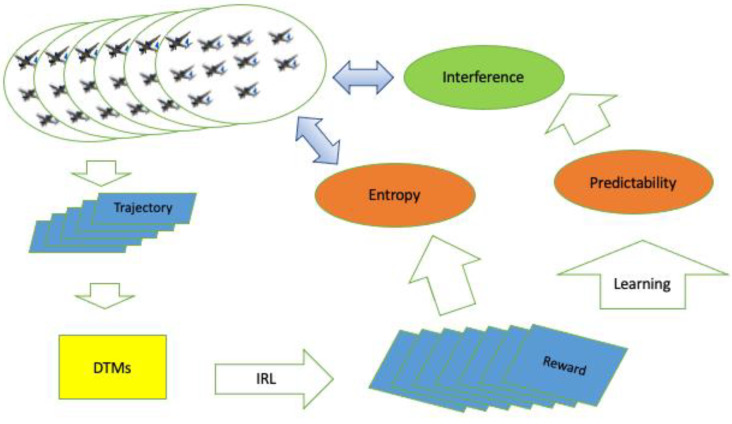
Workflow to estimate entropy and interference.

**Figure 2 entropy-24-01364-f002:**
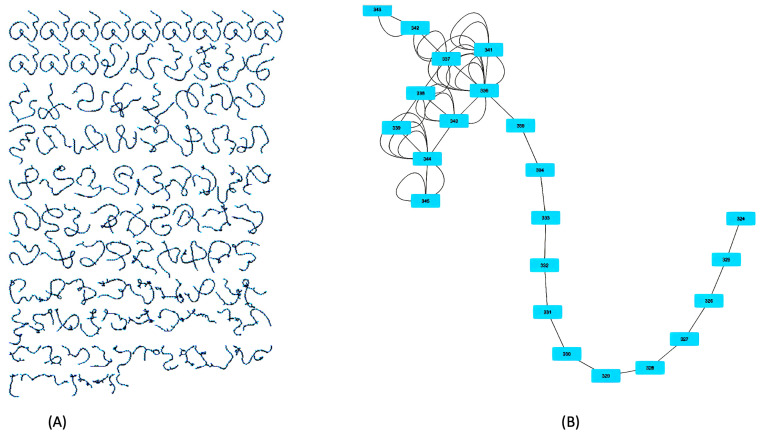
An example DTM. (**A**) Full DTM, (**B**) A part of the DTM.

**Figure 3 entropy-24-01364-f003:**
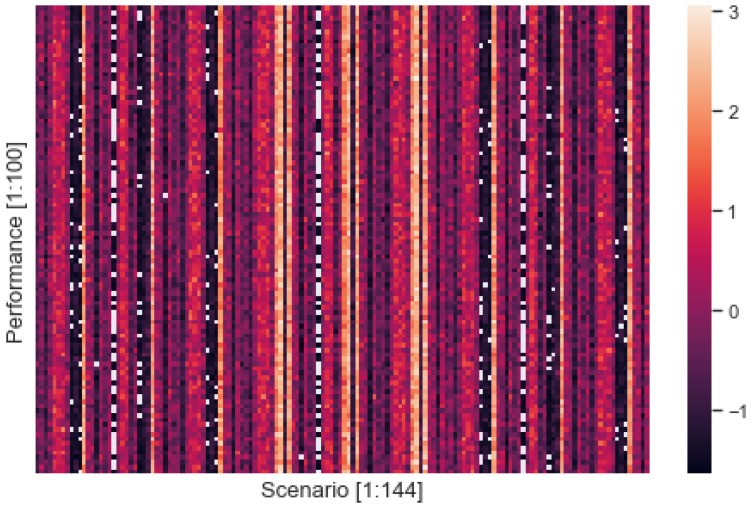
Performance by Scenario.

**Figure 4 entropy-24-01364-f004:**
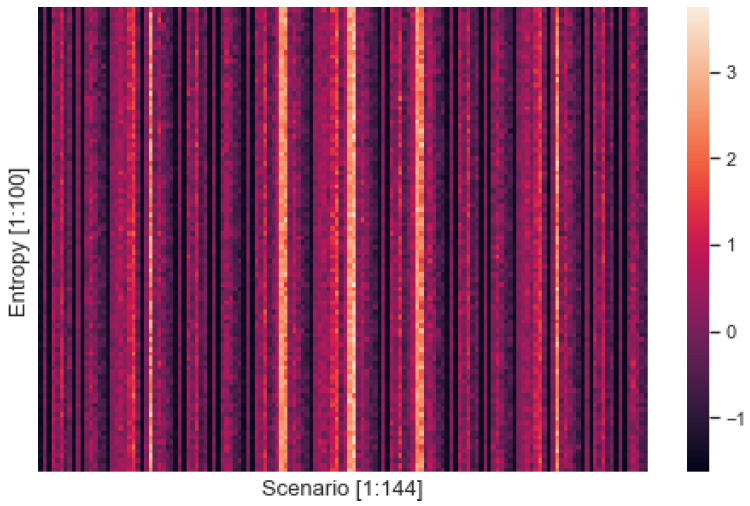
Entropy by scenario.

**Figure 5 entropy-24-01364-f005:**
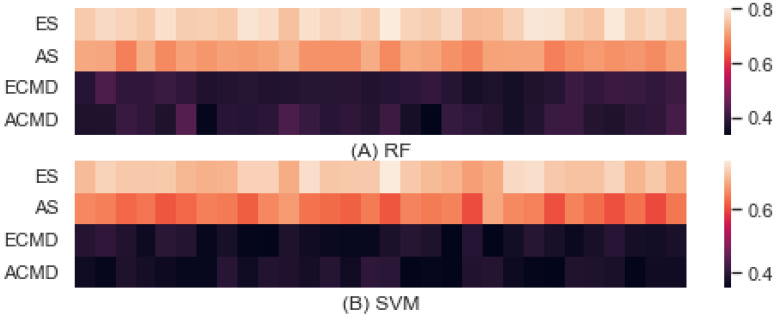
Predictive accuracy by RF and SVM.

**Figure 6 entropy-24-01364-f006:**
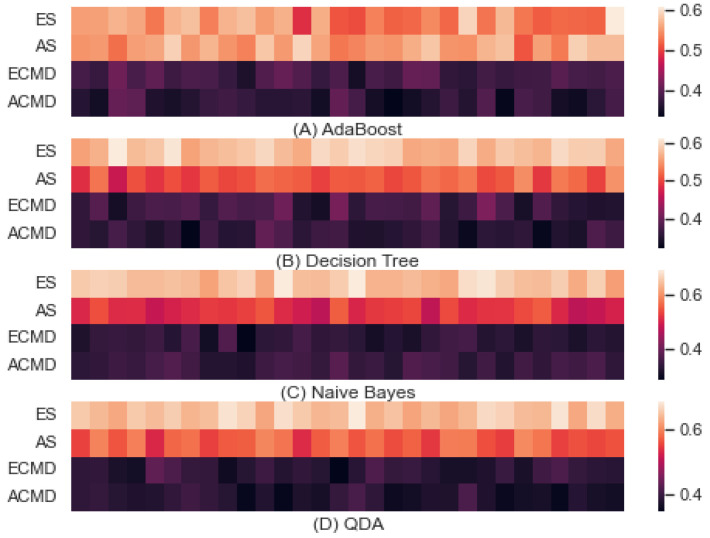
Predictive Accuracy by AdaBoost, Decision Tree, Naive Bayes, and QDA.

**Figure 7 entropy-24-01364-f007:**
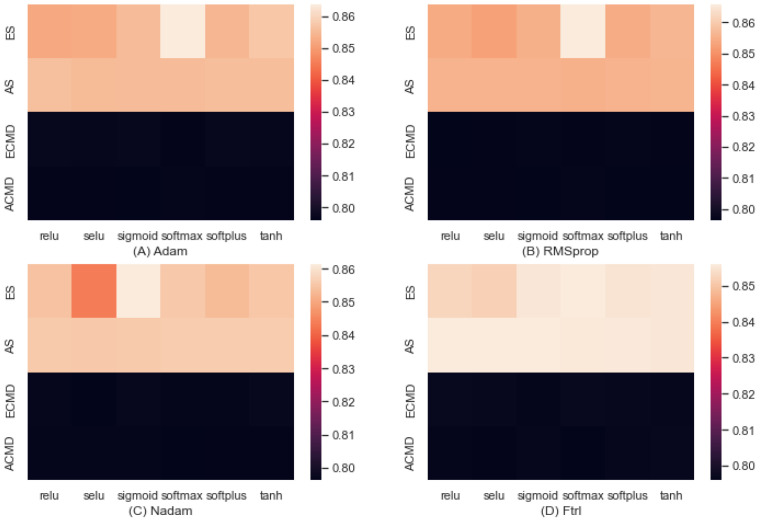
Predictive Accuracy by Deep Learning.

**Figure 8 entropy-24-01364-f008:**
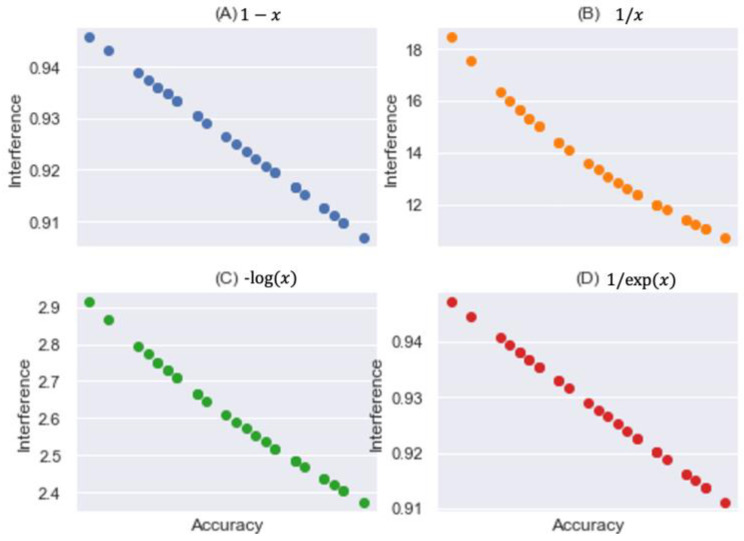
Predictive Accuracy vs. Interference.

**Figure 9 entropy-24-01364-f009:**
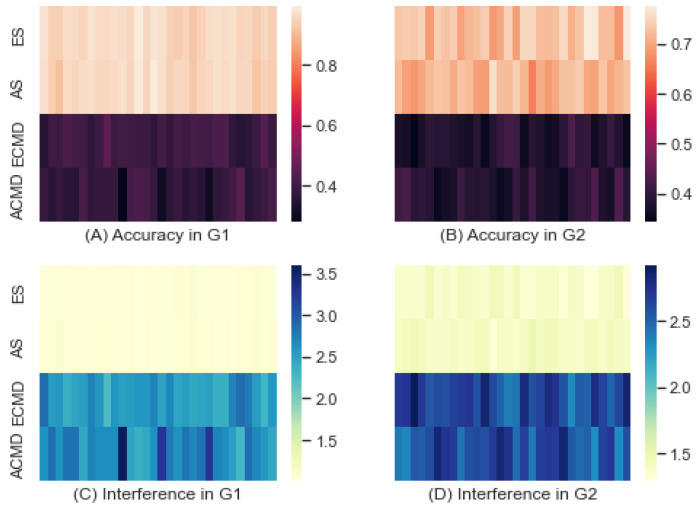
Accuracy and Interference of G1 and G2.

**Figure 10 entropy-24-01364-f010:**
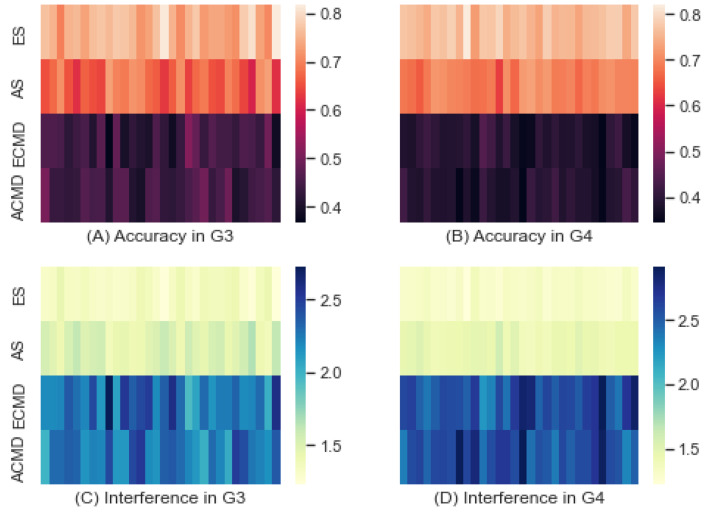
Accuracy and Interference of G3 and G4.

**Figure 11 entropy-24-01364-f011:**
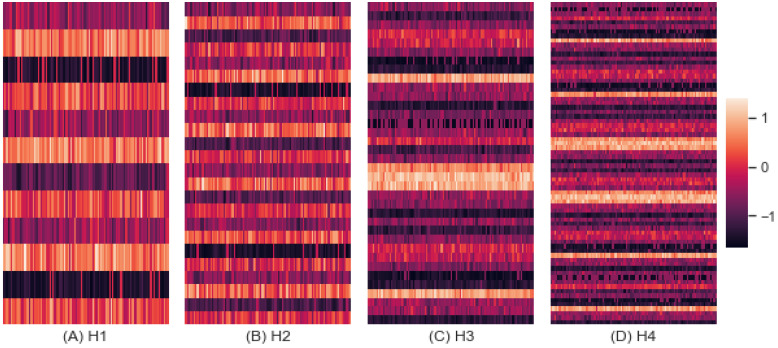
Performance by Homogeneity.

**Figure 12 entropy-24-01364-f012:**
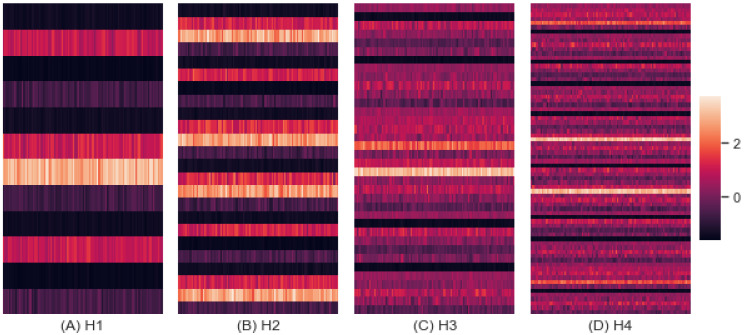
Entropy by Homogeneity.

**Figure 13 entropy-24-01364-f013:**
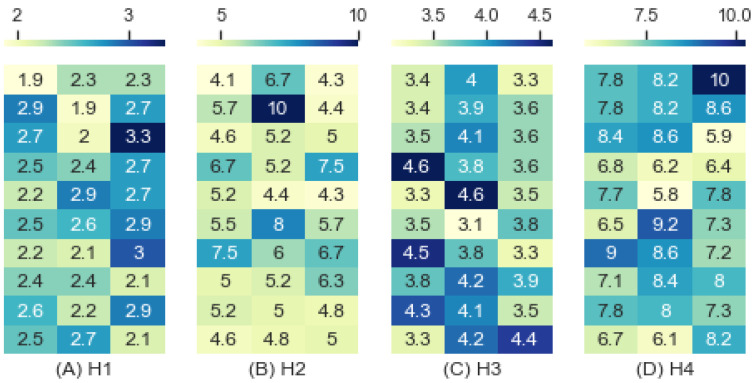
Interference by Homogeneity.

**Figure 14 entropy-24-01364-f014:**
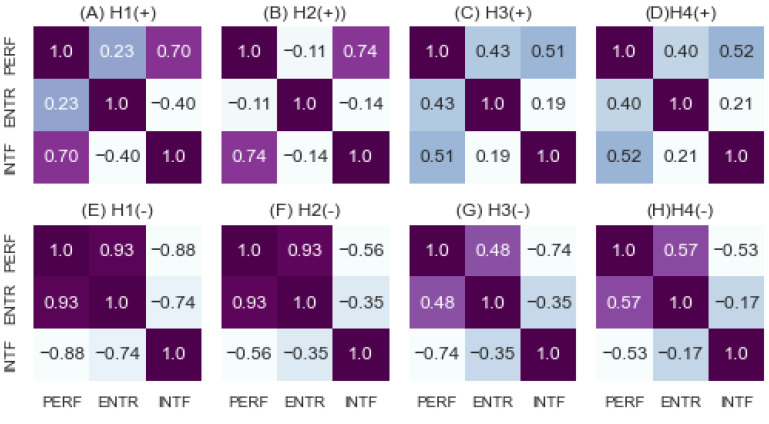
Positive and Negative Correlations.

**Table 1 entropy-24-01364-t001:** Models for Interference.

	Formula	Range	Mean	STD
A	1−x	[0.907: 0.946]	0.925	0.011
B	1/x	[10.746: 18.461]	13.613	2.032
C	−log(x)	[2.375: 2.916]	2.600	0.146
D	1/exp(x)	[0.911: 0.947]	0.928	0.011

**Table 2 entropy-24-01364-t002:** Correlations and *p*-values associated with Figure 12.

H	Metric1	Metric2	Correlation	*p*-Value	# Positive	H	Metric1	Metric2	Correlation	*p*-Value	# Negative
1	PERF	ENTR	0.236	0.702	5	1	PERF	ENTR	0.936	0.002	7
1	PERF	INTF	0.704	0.185	5	1	PERF	INTF	−0.884	0.008	7
1	ENTR	INTF	−0.407	0.496	5	1	ENTR	INTF	−0.743	0.056	7
2	PERF	ENTR	−0.113	0.726	12	2	PERF	ENTR	0.934	<0.001	12
**2**	PERF	INTF	0.745	0.005	12	2	PERF	INTF	−0.567	0.055	12
2	ENTR	INTF	−0.144	0.655	12	2	ENTR	INTF	−0.352	0.261	12
3	PERF	ENTR	0.439	0.068	18	3	PERF	ENTR	0.485	0.042	18
3	PERF	INTF	0.514	0.029	18	3	PERF	INTF	−0.744	<0.001	18
3	ENTR	INTF	0.195	0.439	18	3	ENTR	INTF	−0.351	0.153	18
4	PERF	ENTR	0.402	0.011	39	4	PERF	ENTR	0.576	<0.001	33
**4**	PERF	INTF	0.527	0.001	39	4	PERF	INTF	−0.538	0.001	33
4	ENTR	INTF	0.213	0.194	39	4	ENTR	INTF	−0.172	0.339	33

**Table 3 entropy-24-01364-t003:** Combat Strategy and Command Style Combinations.

Positive Correlations					Negative Correlations				
H	st1	St2	Style1	Style2	H	St1	St2	Style1	Style2
1	BF	BF	SPONT	SPONT	1	CS	CS	INT: RATNL: SPONT	INT: RATNL: SPONT
1	FF	FF	INT: RATNL: SPONT	INT: RATNL: SPONT	1	BF	BF	INT: RATNL	INT: RATNL
1	AN	AN	RATNL	RATNL	1	AN	AN	INT: SPONT	INT: SPONT
2	CS	CS	INT: RATNL: SPONT	RATNL: SPONT	2	CS	CS	INT: RATNL: SPONT	SPONT: INT
2	BF	BF	RATNL	INT	2	BF	BF	INT: RATNL: SPONT	RATNL: SPONT: INT
2	FF	FF	INT: RATNL: SPONT	RATNL: SPONT: INT	2	AN	AN	INT: RATNL: SPONT	RATNL: SPONT: INT
2	AN	AN	RATNL: SPONT	INT: RATNL	3	CS	AN: BF	INT: SPONT	INT: SPONT
3	CS	BF: FF: AN	INT: RATNL: SPONT	INT: RATNL: SPONT	3	BF	CS: FF: AN	INT: RATNL: SPONT	INT: RATNL: SPONT
3	BF	FF	RATNL	RATNL	3	FF	AN: CS: BF	INT: RATNL: SPONT	INT: RATNL: SPONT
3	FF	CS: BF: AN	INT: SPONT	INT: SPONT	3	AN	FF: BF	INT: RATNL	INT: RATNL
3	AN	CS: BF: FF	INT: RATNL: SPONT	INT: RATNL: SPONT	4	CS	AN	RATNL	RATNL
4	CS	BF: FF: AN	INT: RATNL: SPONT	RATNL: SPONT: INT	4	BF	CS: AN: FF	INT: RATNL: SPONT	RATNL: SPONT: INT
4	BF	FF: AN: CS	INT: RATNL: SPONT	RATNL: INT: SPONT	4	FF	AN: CS: BF	INT: RATNL: SPONT	SPONT: INT: RATNL
4	FF	CS: BF: AN	INT: SPONT	RATNL: SPONT: INT	4	AN	CS: BF: FF	INT: RATNL: SPONT	SPONT: INT: RATNL

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
