# Peer review of "An Application of Inverse Reinforcement Learning to Estimate Interference in Drone Swarms"

_entropy, 2022, doi:10.3390/e24101364_

Round 1

Reviewer 1 Report

Please add more references and improve the readability of the paper. 

Could you please use deep learning methods and compare the results with SVM?

A random forest model is used, it would be good if you provide some results based on a decision tree method as well.

Author Response

  1. Please add more references and improve the readability of the paper.

Thank you for this suggestion. We have added more references and rewritten the abstract and discussion to better address the objectives and approaches taken to improve the readability. We added two more figures (i.e., Figure 1 and 2) to improve readability.

  1. Could you please use deep learning methods and compare the results with SVM?

We conducted additional experiments by applying deep learning as suggested and extended the paper to include additional results in section 3.3.1.2. By SVM, we obtained the overall mean accuracy of 0.523 with a standard deviation of 0.1750. By deep learning, we obtained the overall mean accuracy of 0.826 with a standard deviation of 0.0298. Deep learning provided better accuracies with less deviations.

  1. A random forest model is used, it would be good if you provide some results based on a decision tree method as well.

As suggested, we conducted additional experiments by applying decision tree and integrated the results in section 3.3.1.1. Compared with the results by RF that provided the overall mean accuracy of 0.563 with a standard deviation of 0.2037, Decision Tree provided the overall mean accuracy of 0. 4548 with a standard deviation of 0. 1102. RF showed higher accuracies than Decision Tree.

Reviewer 2 Report

In this paper authors monitor movements of drone swarms and use the entropy & interference associated with them to make inferences on the original intent of the drones. They simulate a variety of command styles and combat strategies to validate their result. 

Strong point: 

++The topic is relevant and interesting, the market is ready for this. 

++The main contribution, which is associating predictability with interference, is a no cost solution for a relevant problem.

Weakpoint: 

—Writing needs improvement. 

—The paper lacks scientific value. 

—The paper lacks benchmarking. 

Comments: 

The writing quality is poor and needs to be improved significantly, otherwise the work is not ready to get published. The authors have written the paper as their work progressed which makes it more like a lab report than a journal paper. Sometimes the paper is too wordy and there is a lot of details on intermediate results which has made the authors compromise on high level conclusions and interpretations. Please apply this carefully to the manuscript before submitting a revision. 

The paper title is way too general. 

Here is how the abstract begins: it informs the reader of existing challenges, and jumps to how to tackle them without actually introducing the challenges. Plz update the abstract. 

Introduction includes low level details which can be replaced with high level main contributions and why they are important or how they have been overlooked in the SOTA. 

In the last paragraph of the introduction authors claim to have experimental results in addition to simulations which in my opinion has not happened! 

In the beginning of Section 2, how do you choose the number of drones in the swarm? You need to justify the practicality of this number by adding a reference. 

It would be nice to add graphics with sequence of events to show the application of Markov series in DTM studies. Plz take this as a suggestion. 

The results pertaining to “interference” are counterintuitive and there is no clarification on this in the main body of the paper. “The groups that experience more interference, perform better”!!! Isn’t more interference a corollary of less predictability? Is the ultimate goal of the friend/foe drone swarms to be less predictable? 

In Section 2.3, “Interference leads to observed values being different from the actual values”. A few sentences after this state that it is a double edged sword and could be good or bad. I do not think the writing is well enough so the ppl who are not experts in this field could read and understand the work. 

First paragraph of Section 2.3 could be rephrased to improve its fluency. 

Why is entropy in Eq. 8 representing uncertainty? How does the formula convey this meaning? This is another instant throughout this work that a counterintuitive statement is not elaborated on! 

Where in the paper is the function f(.) in equation 9 defined in mathematical form? If I am correct, g(.) is the predictive accuracy which might not have a closed from. Is that correct? 

In Section 3.1, what is the criteria for winning the game? 

In Section 3.3.1, “The ally side strategy is more predictable than the enemy side for AdaBoost”. Why does this happen? (Please apply this to the entire paper,  such conclusions and interpretations are more important than reporting the numbers).

RF is the chosen method here, how do the authors avoid overfitting as the main drawback of using RF in learning? 

Can figure 5 be used to find a good estimate for the earlier mentioned function f(.)? 

The same figure lacks proper labeling in subfigures A and B. 

What is the upper bound for the interference numbers in figure 10? The color scale on the right does not cover the numbers within the heatmap. For instance, 10 is not within the scale. Plz update the graph and add more intuitive sentences to the caption. 

The paper has many intermediate results. In my opinion Figure 11 could be removed since it is followed by Figure 12 which is the final and last observation. 

Please add a flow graph to show the high level steps of your work including input and output. This could be at the intro or section 2. 

The discussion which is the main part is extremely underdeveloped! 

What are the limitations of the work? What are the future steps? How would this work extend to practical cases?

The paper lacks benchmarking which is very important and cannot be neglected. 

Best of luck!

Author Response

  1. The writing quality is poor and needs to be improved significantly, otherwise the work is not ready to get published. The authors have written the paper as their work progressed which makes it more like a lab report than a journal paper. Sometimes the paper is too wordy and there is a lot of details on intermediate results which has made the authors compromise on high level conclusions and interpretations. Please apply this carefully to the manuscript before submitting a revision.

Thank you for this comment. We have rewritten the abstract and discussion to better address the objectives and approaches taken in this paper. We removed some intermediate results and added some pictures to describe the contents better and to improve readability.

  1. The paper title is way too general.

Tentatively, we changed into “Investigating various drone scenarios using reward distributions driven by Inverse Reinforcement Learning.”

  1. Here is how the abstract begins: it informs the reader of existing challenges, and jumps to how to tackle them without actually introducing the challenges. Plz update the abstract.

Thank you for the comment. We rewrote the abstract as suggested.

  1. Introduction includes low level details which can be replaced with high level main contributions and why they are important or how they have been overlooked in the SOTA.

We changed the introduction to describe high level concepts and contributions as suggested.

  1. In the last paragraph of the introduction authors claim to have experimental results in addition to simulations which in my opinion has not happened!

So far, all our analysis was based on simulations. We corrected the statement.

  1. In the beginning of Section 2, how do you choose the number of drones in the swarm? You need to justify the practicality of this number by adding a reference.

We added the reference1.

  1. It would be nice to add graphics with sequence of events to show the application of Markov series in DTM studies. Plz take this as a suggestion.

Figure 2 was added to show an example DTM.

  1. The results pertaining to “interference” are counterintuitive and there is no clarification on this in the main body of the paper. “The groups that experience more interference, perform better”!!! Isn’t more interference a corollary of less predictability? Is the ultimate goal of the friend/foe drone swarms to be less predictable?

The comment of ‘The results pertaining to “interference” are counterintuitive’ would be based on the intuition that the interference always impacts performance negatively. However, not all interference impacts negatively. You are right that our results showed that the groups experienced more interference, performed better when we investigated it by homogeneity. This was discussed more thoroughly in revision. 

  1. In Section 2.3, “Interference leads to observed values being different from the actual values”. A few sentences after this state that it is a double edged sword and could be good or bad. I do not think the writing is well enough so the ppl who are not experts in this field could read and understand the work.

We rewrote Section 2.3. to make the flow better.

  1. First paragraph of Section 2.3 could be rephrased to improve its fluency.

We absolutely agree with you. Section 2.3. was restructured and rewritten.

  1. Why is entropy in Eq. 8 representing uncertainty? How does the formula convey this meaning? This is another instant throughout this work that a counterintuitive statement is not elaborated on!

Entropy has been used to understand the behavior of stochastic processes since it represents the uncertainty, ambiguity, and disorder of the process17. Assuming reward distributions as Shannon stochastic information source, we tried to compute the uncertainly of behaviors of drone swarms. In revision, this was addressed in Section 2.3.1.

  1. Where in the paper is the function f(.) in equation 9 defined in mathematical form? If I am correct, g(.) is the predictive accuracy which might not have a closed from. Is that correct?

You are right. Table 1 and Figure 8 (in revision) show our process to consider some candidates of f() in Eq (9).

  1. In Section 3.1, what is the criteria for winning the game?

Win or lose the game is determined by either a team’s ship is destroyed, or all drones in the team died. If either of those was not accomplished within given time limit (i.e., 50 steps), whoever has the most (health + (ammo/2)) wins the game. This was added into Section 3.1.

  1. In Section 3.3.1, “The ally side strategy is more predictable than the enemy side for AdaBoost”. Why does this happen? (Please apply this to the entire paper, such conclusions and interpretations are more important than reporting the numbers).

Based on IRL formulations provided in Section 2.2, ally drones were set to target, and enemy drones were set to non-target. The difference between ally and enemy strategy (same as ally and enemy command style) seems a side effect or noise since it was inconsistent in applying various machine learning methods and became negligible when we applied deep learning methods as suggested by reviewers.

  1. RF is the chosen method here, how do the authors avoid overfitting as the main drawback of using RF in learning?

We could not fully investigate to overcome the limitations of RF. RF was one of many applicable learning methods. We simply chose RF due to easy implementation and fast outcome.

  1. Can figure 5 be used to find a good estimate for the earlier mentioned function f(.)?

That is correct.

  1. The same figure lacks proper labeling in subfigures A and B.

All labels were added to Figure 5 (i.e., Figure 8 in revision).

  1. What is the upper bound for the interference numbers in figure 10? The color scale on the right does not cover the numbers within the heatmap. For instance, 10 is not within the scale. Plz update the graph and add more intuitive sentences to the caption.

We set the upper bound of interference with 100. We fixed the figure 10 to display right color bars for each heatmap.  

  1. The paper has many intermediate results. In my opinion Figure 11 could be removed since it is followed by Figure 12 which is the final and last observation.

We agree. We removed the unnecessary figure and contents.

  1. Please add a flow graph to show the high level steps of your work including input and output. This could be at the intro or section 2.

We added Figure 1 to show the high-level workflow of our research.

  1. The discussion which is the main part is extremely underdeveloped!

Discussion was developed thorough as suggested.

  1. What are the limitations of the work? What are the future steps? How would this work extend to practical cases?

As noted in discussion, our analysis was based on simulations that are limited in diversity of combat strategy and command style. So far, only four level of homogeneity has been addressed. We are developing more combat strategies to extend this study.

  1. The paper lacks benchmarking which is very important and cannot be neglected.

The objective of our research is to develop a computational framework to deduce the original intent of drone controllers by monitoring their movements. To the best of our knowledge, there is no right benchmark system to derive human intent from behavior records. In this work, we tried to deduce the intent through estimating interference. As noted in the paper, interference has not been well formulated in general applications. In many cases, interference was measured by reading signals through external devices, which is basically different from our approach and not applicable as benchmark. Therefore, we computed entropy to compare against interference inferred from predictability. Both were computed from reward distributions uncovered by IRL, showed similar patterns when comparing groups by homogeneity, but different quantities and associations in correlation analysis. We presumed that interference could be measured by a probability of unexpected behaviors occurring in drone swarms and capture something different from entropy.    

Round 2

Reviewer 2 Report

The authors have done great work in answering the questions in the previous round of review. Here are some overlooked items that need to be addressed before the work gets ready for publication. 

The title still only shows “how” and does not show “why” 

The added figures are great but are blurry. 

The English writing needs to be improved by an Editor. The quality has improved but is not sufficient yet. 

I am not convinced about using RF and sticking with it. These libraries are already implemented which should make it easy for the authors to try a couple of more methods to see what’s best. The data analysis should be more extensive. Plz put in the work needed. 

Author Response

  1. The title still only shows “how” and does not show “why” 

We appreciate you giving us all those comments. We tried to put “how” and “why” in this title: “An Application of Inverse Reinforcement Learning to Estimate Interference in Drone Swarms.”

  1. The added figures are great but are blurry. 

 We made Fig 2 clearer.

  1. The English writing needs to be improved by an Editor. The quality has improved but is not sufficient yet. 

We revised the paper further.

  1. I am not convinced about using RF and sticking with it. These libraries are already implemented which should make it easy for the authors to try a couple of more methods to see what’s best. The data analysis should be more extensive. Plz put in the work needed. 

For 144 scenarios specified by combining multiple combat strategies and command styles, we built 100 DTMs for each scenario and analyzed them to estimate interference from predictability accomplished by learning method including deep learning as provided. A DTM was built from trajectories collected by 100 simulations for each scenario. RF was neither the best among all learning methods employed, nor the only one applicable to our framework. Deep learning showed higher accuracies than RF. RF was only the learning method chosen arbitrarily. In general, researchers don’t have to be convinced about using RF or stick with it to utilize our approach.  
